# Experimental Study on Water-Plugging Performance of Grouted Concrete Crack

**DOI:** 10.3390/ma17071568

**Published:** 2024-03-29

**Authors:** Lianzhen Zhang, Changxin Huang, Zhipeng Li, Anni Wang, Meng Gao, Yang Gao, Xiaochen Wang

**Affiliations:** 1College of Pipeline and Civil Engineering, China University of Petroleum, Qingdao 266580, China; zhanglianzhen@upc.edu.cn (L.Z.);; 2School of Civil Engineering, Shandong University, Jinan 250061, China; huang_changxin@163.com; 3School of Transportation and Civil Engineering, Shandong Jiaotong University, Jinan 250357, China; 4Key Laboratory of Large Structure Health Monitoring and Control, Shijiazhuang Tiedao University, Shijiazhuang 050043, China; 5School of Civil Engineering, Shandong Jianzhu University, Jinan 250101, China; wangsevenine@163.com

**Keywords:** geotechnical engineering, grouted concrete crack, water plugging capacity, grouting material, microstructure

## Abstract

In this paper, ordinary Portland cement, ultrafine cement, polyurethane, and epoxy resin were selected as typical grouting materials. Grouting simulation tests were first conducted to prepare the grouted concrete crack sample. The effect of concrete crack parameters (i.e., crack aperture and roughness), grout water–cement ratio, and grouting pressure on the water-plugging performance of different grouting materials was explored through the impermeability test. The microstructure of grouted concrete cracks was analyzed by means of scanning electron microscopy (SEM) and computed tomography (CT), and the difference in water-plugging performance of different grouting materials was explained at the micro level. The results show that the impermeability of the four grouting materials was ranked as follows: Epoxy resin > polyurethane > ultra-fine cement > ordinary Portland cement. The concrete cracks grouted by epoxy resin have the highest plugging failure water pressure and the lowest permeability, which is the optimal grouting material. The effectiveness of crack grouting in water-plugging was directly proportional to the grouting pressure, provided the pressure did not exceed a certain value. When the pressure surpassed the threshold, the increase in pressure did not have a significant impact on the water plugging performance. For the two cement-based materials, the threshold pressure was 1 MPa, while for the other two chemical grouts, it was 2 MPa. The two cement-based grouts with a water–cement ratio of 0.8 showed optimal water-plugging performance. The water-plugging performance of ordinary Portland cement paste, ultra-fine cement pastes, and polyurethane grout was negatively correlated with crack aperture and positively correlated with crack roughness. However, the water-plugging performance of epoxy resin grout was not affected by crack aperture or roughness.

## 1. Introduction

Water leakage from cracks in lining concrete is one of the main diseases in operation tunnels [1,2,3]. According to statistics, about 30% of the world’s operational railway and highway tunnels have certain of these problems [4]. Grouting is the method that injects cementitious materials into concrete cracks to fill them, reduce the scale and connectivity of seepage channels, and achieve the purpose of sealing leakage water. As this method has low cost and better performance in sealing water leakage, it has become the main method of sealing water leakage from cracks in tunnel-lining concrete.

With the continuous increase in the number of operation tunnels, the issue of grouting control in tunnel lining water leakage has gradually become a hot research topic [5,6,7,8]. Due to the large-scale tunnel lining, crack structure is difficult in the experimental simulation [9,10]. Most previous research simplifies the lining crack structure into rock cracks, and indirectly determines the behavior of grouting in concrete cracks by investigating the grouting diffusion and sealing laws of rock cracks. In terms of grouting diffusion in rock cracks, Funehag et al. [11] used the self-developed crack model system to explore the relationship between the grouting time and the penetration length in smooth cracks under a constant grouting pressure condition. Draganovic et al. [12,13] used the smooth steel channel to simulate a flat crack and conducted the grouting test. The variation in grout pressure in the position of crack shrinkage was obtained to indirectly evaluate the filtration phenomenon in this position. Lu et al. [14] investigated the evolution of filter cake where grout passed through the crack with varying apertures. The minimum grouting pressure that affects the filter cake was determined. Wang et al. [15] designed a large-scale test system for simulating grouting in rough cracks. The grout concentration distribution along the penetration path was obtained. In terms of grouting sealing in rock cracks, Li et al. [16,17] developed a model test system for simulating flat crack grouting in a dynamic water environment. The grouting sealing law of dynamic water was explored through tests. On this basis, Sui et al. [18] further studied the dynamic water grouting sealing effect of chemical slurry. The main factors affecting the sealing effect of chemical slurry were analyzed. Yang et al. [19,20] used the regular sawtooth shape to simulate the undulations of the crack surface. They studied the grouting sealing law of rough cracks in carbon fiber composite cement under dynamic water conditions. Liu et al. [21] used Barton’s standard JRC profiles to describe the roughness of the crack surface in tests. They investigated the grouting performance of the chemical and cement-based grout in a rough crack with flowing water. However, most existing experimental works on concrete crack grouting failed to consider the water-plugging performance of grouted concrete cracks from the perspective of mechanism. Few studies focus on the anti-seepage performance of grouted concrete cracks, which makes it hard to provide effective guidance for grouting plugging water leakage from lining concrete cracks. Grouting control in water leakage from lining concrete cracks is still at a stage of strong experience and blindness. Grouting control in water leakage from lining concrete cracks is still at an empirical and blind stage. Thus, a better understanding of the grouting plugging mechanism in concrete cracks is important.

In this paper, we selected ordinary Portland cement, ultrafine cement, polyurethane, and epoxy resin as the typical grouting material. The permeability and failure water pressure of concrete cracks after grouting were used as evaluation indicators. Based on the concrete crack grouting simulation test system and impermeability test system, the influence of grout water–cement ratio (W/C), grouting pressure, crack aperture, and roughness on the water-plugging performance of grouting in concrete cracks was investigated. The microstructure of concrete cracks after grouting was analyzed by means of SEM and CT methods. The difference in crack water-plugging performance for different grouting materials was explained at the micro level. The research results will provide guidance in selecting suitable grouting materials and parameters according to the crack property (aperture, roughness) in engineering practice.

## 2. Experimental Apparatus, Materials and Scheme

### 2.1. Experimental Apparatus

#### 2.1.1. Grouting Simulation Test System

As shown in Figure 1, the grouting simulation test system for concrete cracks consists of a test frame, injection machine, and supporting pipelines. It can simultaneously realize the concrete crack sample fix, grouting process control, and visual observation.

The grouting machine, which is equipped with a pressure gauge to observe the grouting pressure, is used as the power source. The grouting machine is connected to the grouting needles by a high-pressure hose for the delivery of the grout. The grouting needle is placed in the grouting hole of the concrete crack sample. The grouting hole is vertically connected to the crack. The grouting needle uses an expansion rubber ring to ensure sealing between the grouting needle and the side wall of the grouting hole. The grout is injected into the concrete cracks through the grouting needle, and the test frame observation port valve is opened and closed to observe the grouting process. A one-way valve structure is installed inside the grouting needle to ensure that the grout can only flow into the crack from the grouting machine and not flow back.

The test frame of the grouting simulation is shown in Figure 2. It mainly consists of an upper plate, lower plate, baffle, observation valve, and connecting rod. The upper plate and lower plate are connected through a connecting rod. The upper and lower sides of these plates are provided with fastening nuts, where the nuts on both sides strictly limit the displacement of these plates in the upper and lower directions. The position of the concrete crack sample and crack aperture can maximally remain constant during the test.

#### 2.1.2. Impermeability Test System

The impermeability test system is shown in Figure 3. This system can investigate the permeability and seepage failure pressure of grouted concrete crack samples. The system contains a sealed cabin, which consists of a steel tube and water bladders. By applying water pressure to the water bladder, confining pressure can be applied to the tested sample to prevent the formation of seepage channels on the side wall of the sample during the permeability test. The accuracy of the permeability test results can be ensured.

During the permeability test, water pressure is applied to one end of the sample, and the other end is the free outflow end. The confining pressure of the sample maintains a pressure difference of 2 MPa from the water pressure at the pressurized end of the sample. Set the initial water pressure to 0.1 MPa, and increase it by 0.1 MPa every 1 h. Observe whether there is water seepage from the sealed cabin outlet, and use this as a criterion to determine the seepage failure pressure of grouted concrete crack samples. Based on the water pressure difference on both sides of the sample, the stable seepage velocity, and the size of the sample, Darcy’s law was used to calculate the permeability of the grouted concrete crack samples.

### 2.2. Materials and Concrete Crack Sample

#### 2.2.1. Grouting Material

The commonly used grouting materials in the tunnel project, including ordinary Portland cement, ultrafine cement, polyurethane, and epoxy resin, were selected as the typical grouting material for the experiment of grouting sealing in concrete cracks (see Figure 4). The detailed parameters of the grouting material are shown in Table 1 and Table 2. Among them, mesh number, a general standard for characterizing the material particle size, was used here to represent the size property of ultrafine cement. The strength parameter was obtained following ASTM C942-15 [22] and ASTM C78-09 [23].

#### 2.2.2. Grouted Concrete Crack Sample

Grouted concrete crack samples of different types of material were prepared in two steps. The first step was to prepare concrete cracks, and the second step was to make a grouted concrete crack sample.

As shown in Figure 5, the preparation process of the concrete crack sample with different JRC was as follows: Joint roughness coefficient (JRC) was used to characterize the roughness of the concrete crack [24]. Two JRC, respectively JRC = 6~8 and JRC = 18~20, were selected to describe the relatively smooth and rough morphology of the crack. The steel block with the JRC profile was used as the stencil. A single concrete crack with roughness morphology was obtained by pouring cement mortar made from ordinary Portland cement PO42.5 onto steel blocks. Steel pieces with thicknesses of 1 mm, 2 mm, and 3 mm were pasted on the four corner points of the concrete crack. Two concrete cracks with the same roughness morphology were put together, and a plate was pasted on the side of the concrete crack. A concrete crack sample with different JRC and apertures was obtained.

After the preparation of the concrete crack sample, the grouting machine is used to inject the grout to obtain the grouted concrete crack sample. Demolding was carried out 1 day after grouting and cured for 7 days. After curing completion, core the grouted concrete crack. The center line of the core is located on the overall extension surface of the crack and perpendicular to the bottom surface of the grouted concrete crack. The cylindrical grouted concrete crack is shown in Figure 6. The sample height is 100 mm, and the diameter is 50 mm, which is used to carry out subsequent impermeability tests.

### 2.3. Experimental Scheme

Factors affecting the sealing performance of grouting include crack structural characteristics, grouting materials, and parameters. In this paper, crack aperture and roughness were selected as indicators of crack structure characteristics. The water–cement ratio and the material type were selected as the grout martial factor. An experimental scheme for evaluating the sealing performance of different grouting material was designed.

According to the investigation statistics of water leakage diseases in tunnel lining cracks [25], more than 90% of the tunnel lining cracks with water leakage diseases are open cracks, with a range of crack apertures from 1~3 mm. Thus, the crack aperture was respectively set at 1 mm, 2 mm, and 3 mm. Two JRC, JRC = 6~8 and JRC = 18~20, respectively, were selected to describe the relatively smooth and rough morphology of the crack. The range of grouting pressure was set from 0.5 to 3 MPa, and the grouting pressure gradient was set at 0.5 MPa. The water–cement ratio range of ordinary Portland cement and ultrafine cement was set from 0.6 to 1.4, and the water–cement ratio gradient is 0.2. The test details are listed in Table 3.

## 3. Results and Discussion

### 3.1. Influence of Grouting Pressure on Plugging Capacity

The influence of grouting pressure on plugging capacity under the condition of crack aperture at 1 mm, JRC = 18~20, and W/C = 0.8 is shown in Figure 7.

As displayed in Figure 7, seepage failure pressure was positively correlated with the grouting pressure, and the permeability was negatively correlated with the grouting pressure for the four types of grouts. The water-plugging performance of grouted concrete cracks was mainly related to whether the grout filling was sufficient. When the grouting pressure was small, the grout could not fully diffuse and fill the concrete crack (see Figure 8). It resulted in the existence of unfilled areas and more connected seepage channels inside the concrete crack. So, the water-plugging performance was poor. In practice, the grouting pressure should be increased appropriately to improve the degree of grout filling inside the concrete cracks.

As the grouting pressure increased, the grout filling degree inside the cracks was effectively improved. However, the impermeability of grouted concrete cracks showed obvious phase characteristics as the grouting pressure increased. When the grouting pressure exceeds 1 MPa, the seepage failure pressure of the cement-based grout reaches its peak value. Its permeability reached the minimum value and remained constant. When the grouting pressure exceeds 2 MPa, the seepage failure pressure of the two chemical materials reaches its peak value. Its permeability also reached the minimum value and remained constant. It indicates that when the grouting pressure reaches a certain value, the grout filling degree inside the concrete crack reaches its peak value, and it is hard to improve the water-plugging performance by increasing the grout filling degree. Compared with cement-based grout, chemical-based grout has a higher viscosity. Therefore, it needs a higher grouting pressure to fully fill the concrete crack with chemical-based grout.

Under the same grouting pressure conditions, the water-plugging performance of different grouting material from high to low is: Epoxy resin > polyurethane > ultrafine cement > ordinary Portland cement. The seepage failure pressure of grouted concrete cracks with epoxy resin was 3.95 MPa, which is 23.2, 15.2, and 2.6 times higher than the ordinary Portland cement, ultrafine cement paste, and polyurethane grout conditions, respectively. The permeability of grouted concrete cracks with epoxy resin was 6, 6, and 3 orders of magnitude lower than that of ordinary Portland cement, ultrafine cement, and polyurethane grout, respectively. It showed a better water-plugging performance. The reason may be attributed to two aspects: ① In cement-based grout, in the filling process, there is a bleeding phenomenon and a shrinkage reaction during the condensation process. Especially the shrinkage reaction, which easily breaks the bonding between the cement paste and the concrete crack. Compared with cement-based grout, chemical-based grout does not have a shrinkage reaction and is easier to fill concrete cracks. So, the water-plugging performance of chemical-based grout is better than that of cement-based grout. ② Epoxy resin grout is more compatible with concrete crack sidewalls than polyurethane grout. Besides, the epoxy resin-concrete interface transition zone’s size is larger, making it hard to become the seepage channel. Thus, the water-plugging performance of epoxy resin grout is better than that of polyurethane grout.

### 3.2. Influence of W/C on Plugging Capacity

The influence of the water–cement ratio on plugging capacity under the conditions of crack aperture at 1 mm, JRC = 18~20, and grouting pressure of 1 MPa is shown in Figure 9.

When the water–cement ratio is in the range of 0.6 to 0.8, as the water–cement ratio increased, the water-plugging performance of cement-based grout improved. As the water–cement ratio increased, the seepage failure pressure of grouted concrete cracks with ordinary Portland cement increased from 0.15 MPa to 0.17 MPa, an increase of 13.3%. The permeability of it decreased from 0.7 md to 0.66 md, a decrease of 5.7%. The seepage failure pressure of grouted concrete cracks with ultrafine cement increased from 0.23 MPa to 0.26 MPa, an increase of 13%. The permeability of it decreased from 0.25 md to 0.25 md, a decrease of 10.7%.

When the water–cement ratio ranges from 0.8 to 1.4, the water-plugging performance of ordinary Portland cement and ultrafine cement paste decreases as the water–cement ratio increases. The seepage failure pressure was negatively correlated with the water–cement ratio, and the permeability was positively correlated with the water–cement ratio. The water-plugging performance of cement-based grout reached its peak at a water–cement ratio of 0.8. The grouted concrete crack with ordinary Portland cement was 0.17 MPa, and the permeability was 0.66 md. The grouted concrete crack with ultrafine cement was 0.26 MPa, and the permeability was 0.25 md. When the water–cement ratio was increased to 1.4, the seepage failure pressure of grouted concrete cracks with ordinary Portland cement decreased to 0.1 MPa, which was 41.2% lower than the peak value. The permeability of it increased to 1.7 md, which was nearly 1.5 times higher. The seepage failure pressure of grouted concrete cracks with ultrafine cement decreased to 0.19 MPa, which was 24.0% lower than the peak value. The permeability of it increased to 1.7 md. The main reason is that the water–cement ratio corresponding to the complete hydration reaction of cement materials is different. This water–cement ratio is 0.43 for ordinary Portland cement and 0.5 for ultra-fine cement. When the water–cement ratio is in a small range, the free water in the grout system is completely involved in the hydration reaction, and there is no excess free water to transport cement particles. So, the grout fluidity is poor. It is easy to aggregate and causes blockage in the cracks, resulting in a low degree of filling in the crack and poor impermeability. When the water–cement ratio of grout exceeds its water–cement ratio for the complete hydration reaction, it means that more free water in the grout system is involved in transporting cement particles. The fluidity and the filling degree of grout in the crack are enhanced. However, when the water–cement ratio exceeds the optimal water-plugging ratio, the excess water that is not involved in the hydration reaction is more likely to cause the bleeding phenomenon (see Figure 10). It results in the grout not being able to be well cemented with the concrete crack and generates the seepage channel. Moreover, the specific surface of cement material is large, which is also a key factor for its poor fluidity, especially for ultra-fine cement.

Similar results were reported in [20]. They investigated the diffusion efficiency of carbon fiber composite grouts in rough cracks with flowing water. They concluded that grouting diffusion distance increased with a decrease in fracture aperture, water–cement ratio, and additive concentration. The higher the diffusion distance, the stronger the plugging capacity. Its results on plugging capacity are basically consistent with our research. Compared with Yang’s report, our study further supplements the permeability characteristics of concrete cracks after grouting sealing.

### 3.3. Influence of Crack Aperture on Plugging Capacity

The influence of crack aperture on plugging capacity under cement-based grout and chemical-based grout is shown in Figure 11.

For the grouted concrete crack with diffident grouting material, the crack aperture was significantly negatively correlated with the water-plugging performance. As the crack aperture increased, the seepage failure pressure of grouted concrete cracks decreased, and the permeability increased. The crack aperture increased from 1 mm to 3 mm, the seepage failure pressure of ordinary Portland cement, ultrafine cement, polyurethane, and epoxy resin grouted concrete cracks, respectively, dropped from 0.17 MPa, 0.26 MPa, and 1.52 MPa to 0.14 MPa, 0.21 MPa, and 0.31 MPa. The permeability, respectively, increased from 0.66 md, 0.28 md, and 1.89 × 10^−4^ md to 0.8 md, 0.36 md, and 2.76 × 10^−2^ md. Due to the bleeding phenomenon of ordinary Portland cement and ultrafine cement paste, the seepage channels caused by the bleeding are formed inside the crack. When the crack aperture increased, the size of the seepage channel increased, resulting in a negative correlation between the crack aperture and water-plugging performance. The increase in crack aperture caused an obvious decline in water-plugging performance for the grouted concrete crack with polyurethane grout. The permeability of it increased by two orders of magnitude, and the seepage failure pressure decreased to about 20% of the initial value. The main reason may be that the polyurethane grout reacts with air and water, which generates the porous structure. Between the pores is a grout-hardened layer. Water pressure can break the hardened layer, making the internal pores connected to each other and then forming seepage channels. As the crack aperture increases, the number of pores increases. At the same time, the hardened layer that can be broken through by water pressure also increases accordingly, resulting in poor water-plugging performance.

The water-plugging performance of epoxy resin grout was less affected by the crack aperture. Variation of crack aperture hardly caused the changes in permeability and seepage failure pressure of grouted concrete cracks. As the crack aperture increased from 1 mm to 3 mm, the corresponding changes in the permeability and seepage failure pressure of grouted concrete crack was within 3%. The main reason is that the shrinkage of epoxy resin during the consolidation is small, which helps to improve the adhesion strength between the gel and the concrete crack surface [26,27,28]. In addition, the epoxy resin grout has good chemical stability and can fully fill the tiny pores, which makes it behave with better water-plugging performance.

### 3.4. Influence of Crack Roughness on Plugging Capacity

The influence of crack roughness on plugging capacity under cement-based grout and chemical-based grout is shown in Figure 12, among which the crack aperture of these two materials is 1 mm, the water–cement ratio of cement-based grout is 0.8, and the grouting pressure of cement-based grout and chemical-based grout, respectively, is 1 MPa and 2 MPa.

For cement slurries and ultrafine cement paste, the water-plugging performance of the relatively smooth crack was poorer than that of the relatively rough crack. As the crack roughness decreased from JRC = 18~20 to JRC = 6~8, the seepage failure pressure of ordinary Portland cement and ultrafine cement decreased from 0.17 MPa and 0.27 MPa to 0.14 MPa and 0.22 MPa. The seepage failure pressure decreased by about 18%. The permeability, respectively, increased from 0.65 md and 0.23 md to 1.1 md and 0.38 md and increased by 65~70%. Compared with the crack structure of JRC = 6~8, the crack structure of JRC = 18~20 has larger tortuous fluctuations and longer seepage paths. For cement-based grout, the grout was easy to form a “rock bridge” in the cracks with large undulation, and the “rock bridge” and gel together bear the water pressure, so as to enhance its water-plugging performance.

For polyurethane grout, the crack roughness decreased from JRC = 18~20 to JRC = 6~8, and the permeability of the grouted concrete crack increased from 1.88 × 10^−4^ md to 2.09 × 10^−4^ md, an increase of 11.2%. The seepage failure pressure decreased from 1.53 MPa to 1.49 MPa, a decrease of 2.6%. Crack roughness affects the water-plugging performance of polyurethane grout. An increase in crack roughness leads to the enhancement of water-plugging performance, but to a lesser extent.

The water-plugging performance of epoxy resin grout was almost not affected by the crack roughness. The seepage failure pressure of grouted concrete cracks was maintained at 3.95 MPa, and the permeability was maintained at about 1.82 × 10^−7^ md. The filling of epoxy resin grout inside the crack structure was sufficient under the grouting pressure of 2 MPa. As the epoxy resin grout has excellent water-plugging performance, its impermeability is higher than that of the concrete. The bonding strength with the concrete crack surface is higher, resulting in the water-plugging performance of grouted concrete cracks being maintained at a high level.

### 3.5. Microstructure Property of Grouted Concrete Crack

#### 3.5.1. SEM Analysis

The Brazilian Test is a laboratory test for indirectly determining the tensile strength of rocks, which is also used to obtain the concrete crack sample [26,27]. In this SEM analysis, the Brazilian splitting method was first used to separate the grouted concrete crack along the crack surface and obtain the concrete crack surface and grouted body surface. The failure mode of grouted concrete cracks with cement-based grout was the damage of the interface cementation between the grout gel and the concrete crack. The failure mode of chemical materials is the damage to the grout’s internal structure, but the interface between the grout gel and concrete crack was well bonded. A SEM test was conducted to observe the grout gel bonded with the concrete crack and obtain the morphological structure of the grout gel, distribution characteristics, contact state between particles, and bonding form between the grout gel and concrete crack.

The microstructure magnified 200 times of the grout gel for different grouting materials is shown in Figure 13. As displayed in Figure 13a, ordinary Portland cement gel was not completely bonded to the crack surface. The grout gel exhibited the brittle failure characteristic, and the damaged cemented surface was relatively smooth. The distribution of grout gel was not uniform, and many unfilled areas were inside the crack. When the unfilled areas are interconnected, seepage channels are formed, which seriously affect the impermeability properties of grout gel. As displayed in Figure 13b, the degree of bonding between the ultrafine cement paste gel and the crack surface was high. The grout gel exhibited brittle failure characteristic, and the damaged cemented surface was relatively smooth. The distribution of grout gel in concrete cracks was relatively uniform, but unfilled voids still existed. The size of these unfilled voids and their connectivity were relatively small compared with ordinary Portland cement gel. As displayed in Figure 13c, the polyurethane gel exhibited a porous structure and showed brittle failure. The damaged cemented surface was relatively rough, and the distribution of gel was not uniform inside the concrete crack. The unfilled voids existed inside the crack, and there was no connectivity between voids. As displayed in Figure 13d, the distribution of epoxy resin gel was uniform in the concrete crack and exhibited plastic failure. The gel showed an obvious sign of being pulled and torn, but it was still tightly bonded to the crack surface. There were no obvious voids inside the gel.

Compared with ordinary Portland cement, the distribution of ultrafine cement is more evenly distributed in the concrete crack and more tightly cemented to the crack surface. There are also fewer pores in the grout gel, resulting in better impermeability. Compared with ultra-fine cement, polyurethane has more pores inside the gel, but these pores are not interconnected. The bonding strength between the polyurethane gel and the concrete crack was also higher than that of the ultrafine cement gel. So, its impermeability is better. Epoxy resin grout gel was distributed evenly inside the concrete crack and tightly bonded to the crack surface. There are no voids inside the gel. Thus, it has the most excellent impermeability among the four materials.

The microstructure magnified 20,000 times of the grout gel for different grouting materials is shown in Figure 14. As shown in Figure 14a–c, the gel particles formed, respectively, by the three materials, ordinary Portland cement, ultrafine cement, and polyurethane, were stacked on each other. The shapes were mostly in block form, with many heap aggregates and some outer particles. The aggregates were bonded to each other to form a massive solid matrix structure, and the outer particles were free outside the matrix structure and had no contact with each other. As shown in Figure 14d, the polyurethane gel particles were relatively dispersed among the particles, and the aggregates were relatively discrete, without forming a massive matrix structure.

The particle sizes of ordinary Portland cement gel particles vary greatly, and a large number of aggregates indicate that they have higher strength. However, there is a “face-to-face contact” connection between the aggregates. This connection is relatively loose and prone to relative displacement under the function of external forces, which reflects the characteristics of its brittle failure. Compared with ordinary Portland cement, the particle size of ultrafine cement gel is uniform, and its gel is denser. Ultra-fine cement particles are wrapped in a layer of cement hydration material, making the particles more tightly bonded. The connection takes the form of the “face-to-face contact” connection and the “cementation” connection. However, it can be seen from Figure 14b that there are many micro-cracks between the connections of ultra-fine cement particles. It reflects the dry shrinkage of ultrafine cement materials after solidification, which affects the cementation between particles. The polyurethane gel particles are spherical in shape and have a more uniform particle size. The particles are “cemented” connected. This direct contact connection means that some substance outside the particle unit is difficult to observe under an electron microscope that acts as a cement (maybe an electric charge, a water film, or a glue film). The gel particle is prone to relative displacement under the action of external forces, and its structural strength is poor. The epoxy resin gel particles are very tightly bonded, and the gel is very dense. No free epoxy resin particles can be observed, and the connection between particles is a “cementation-inlay” connection, reflecting its high strength and impermeability properties.

The microstructure magnified 100,000 times of the grout gel for different grouting materials is shown in Figure 15. The particles in the gel of cement-based grout appear in the form of scales. There are fibrous hydrates between the particles, which can increase the bonding strength. When the cement-based grout shrinks or bleeds, the bond between the hydrate and the concrete crack surface breaks down, leading to seepage failure. The polyurethane material particles are spherical particles with a small particle size, which can fill the pores on the crack surface and increase the contact area between the grout and the concrete crack side wall. Epoxy resin grout particles are embedded and bonded to each other. The particles are small and can be filled into the pores of the concrete crack surface. The gel and the crack surface are completely bonded and condensed together. The bonding integrity between the particles is better.

#### 3.5.2. CT Analysis

Since the physical properties of the chemical-based grout are largely different from those of the cement paste, it is hard to use the grayscale process method to reconstruct the three-dimensional model of this grout. This section established a three-dimensional structural model of the grouted concrete crack with cement-based paste based on CT technology. The microstructure of the grouted concrete crack, including the filling condition and connection characteristic of the seepage channel inside the concrete crack, was investigated. The microstructure of grouted concrete cracks with cement-based paste is shown in Figure 16. The blue part represents the unfilled area, and the white part represents the filled area.

The grout gel cannot completely fill the entire crack space, mainly concentrated around the grout inlet. The degree of grout filling decreased as the distance from the grout inlet increased, affecting water-plugging performance. The filling degree of ultrafine cement was significantly higher than that of ordinary Portland cement. It suggests that the water-plugging performance of ultrafine cement is significantly higher than that of ordinary Portland cement. The tortuosity of rough cracks significantly affects the spatial distribution of grout gel, where extreme cases such as concentrated accumulation of grout or no grout filling are likely to occur due to fluctuations in the JRC curve. The crack extension direction smoothly varied at position ① where the grout was completely unfilled, as shown in Figure 16. It is possible that this area accumulates bleeding, leading to the development of an area without grout filling. In Figure 16, the grout can be seen as a “rock bridge” at position ②. The grout was concentrated at the zigzag, and there were sudden changes in the crack. However, it becomes difficult for the grout to reach the outer areas of the rock bridge, resulting in no grout filling in the outer crack area. Moreover, the “rock bridge” has a limited size, which makes it difficult to withstand high water pressure. Hence, it can easily be destroyed and turn into a water conduction channel when water pressure increases.

The research aim of this paper is to investigate the effect of different grouting materials, including cement and chemical materials, on the water-plugging performance of grouted concrete cracks. Assess the key factors affecting the water-plugging performance of grouted concrete cracks from a macro perspective and explain the differences in water-plugging performance of different grouting materials from a micro perspective. Since tunnel engineering is often situated in a water-rich environment, concrete cracks are often exposed to water intrusion environments. Under the long-term seepage erosion of groundwater, the interface between the grouted concrete cracks and slurry stone will deteriorate to a certain extent. However, this paper is not involved in this issue. This issue is engaged in the long-term stability of the grouted concrete crack interface, which is significant for evaluating whether different types of grouting materials can meet the long-term service performance of concrete grouting repair structures. It is also an important direction for future research, which we will carry out in follow-up research.

## 4. Conclusions

The concrete cracks grouted with epoxy resin had the highest seepage failure pressure and the lowest permeability, so their performance was the best. The applicability of the four grouting materials was ranked as follows: epoxy resin > polyurethane > ultrafine cement > ordinary Portland cement.When the grouting pressure does not exceed a certain value, the water-plugging performance of grouted concrete cracks is positively correlated with the grouting pressure. Otherwise, the increase in grouting pressure had no significant impact on the water-plugging performance of grouted concrete cracks. For ordinary Portland cement and ultrafine cement paste, this pressure value was 1 MPa. For polyurethane and epoxy resin grout, this pressure value was 2 MPa.There is an optimal water–cement ratio that makes the cement-based grout have the best water-plugging performance. The optimal water–cement ratios of ordinary Portland cement and ultrafine cement are both around 0.8. When the water–cement ratio of the grout was lower than 0.8, the fluidity of the grout was poor, which caused a low degree of crack filling. When the water–cement ratio of the grout was higher than 0.8, the grout had a bleeding phenomenon, which resulted in the seepage channel between the grout gel and crack side wall.The water-plugging performance of ordinary Portland cement, ultrafine cement, and polyurethane had a negative correlation with the crack aperture and a positive correlation with the crack roughness. The increase in crack aperture was conducive to generating new seepage channels, causing the poor water-plugging performance of grouted concrete cracks. The water-plugging performance of epoxy resin grout was not sensitive to crack aperture or roughness.From the perspective of microstructure, the filling degree of cement-based grout was low compared with chemical-based grout, such as polyurethane and epoxy resin grout. In addition, the filling degree of cement-based grout was affected by grout bleeding, shrinkage, and particle size. Polyurethane grout had many internal pores and was prone to brittle failure, resulting in interconnected pores. Epoxy resin grout has strong structural integrity and good compatibility with the concrete interface, which results in the water-plugging performance of polyurethane grout being lower than that of epoxy resin grout.

## Figures and Tables

**Figure 1 materials-17-01568-f001:**
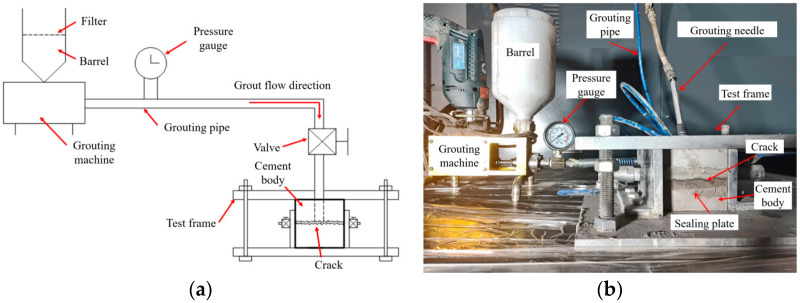
Grouting simulation system: (**a**) schematic diagram; (**b**) experimental system diagram.

**Figure 2 materials-17-01568-f002:**
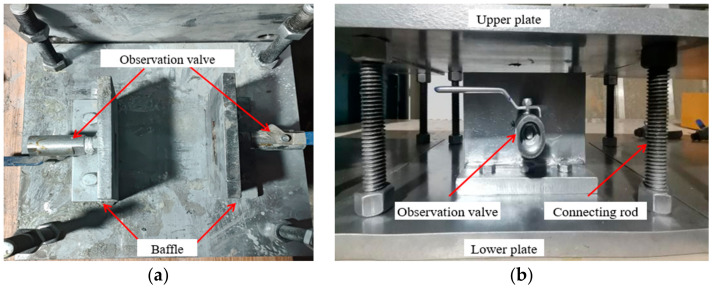
Grouting simulation test frame: (**a**) vertical view; (**b**) front view.

**Figure 3 materials-17-01568-f003:**
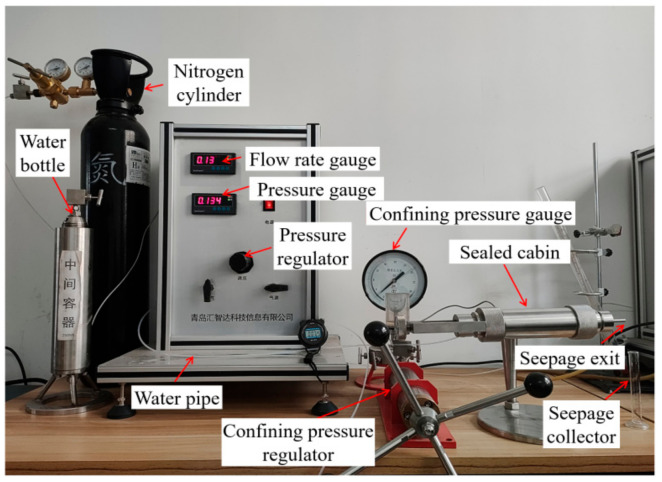
Impermeability test device.

**Figure 4 materials-17-01568-f004:**
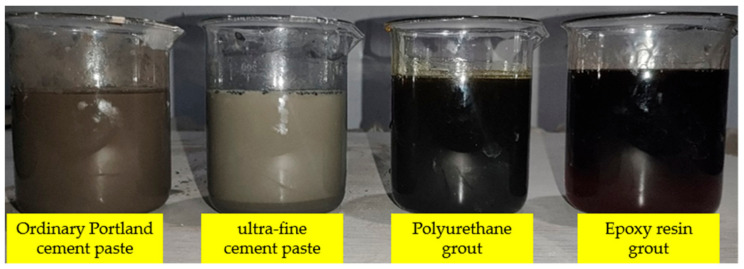
Grouting material.

**Figure 5 materials-17-01568-f005:**
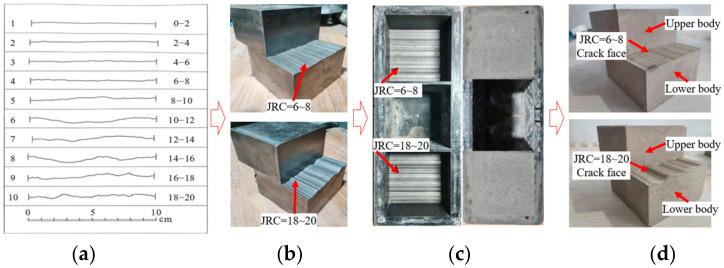
Preparation process of concrete crack sample with different JRC: (**a**) JRC profile; (**b**) steel body; (**c**) cement body; (**d**) cement body with crack.

**Figure 6 materials-17-01568-f006:**
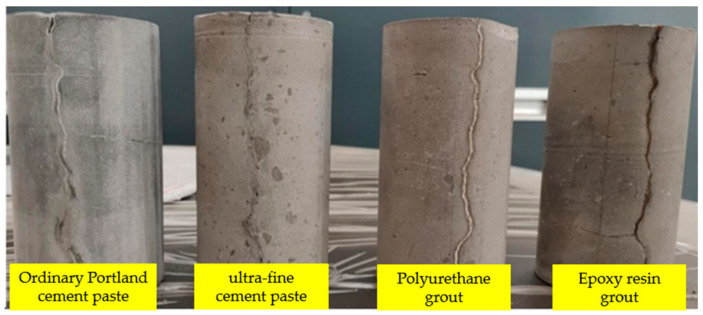
Grouted cylindrical sample.

**Figure 7 materials-17-01568-f007:**
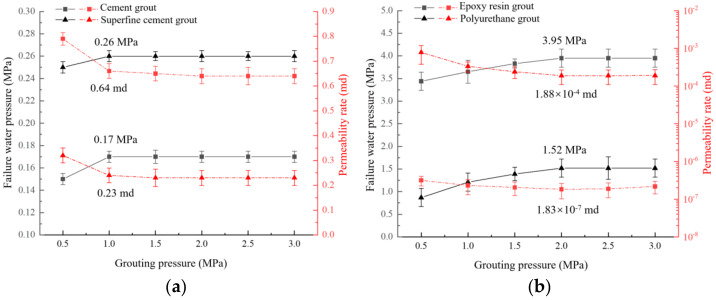
Influence of grouting pressure on plugging capacity: (**a**) cement-based grout; (**b**) chemical grout.

**Figure 8 materials-17-01568-f008:**
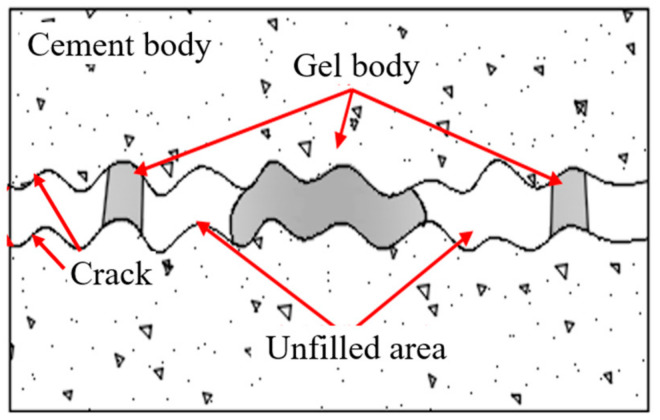
Filling degree in concrete crack.

**Figure 9 materials-17-01568-f009:**
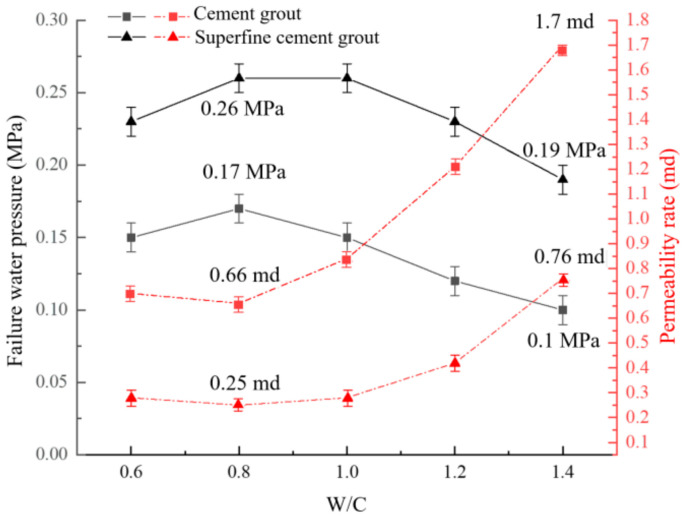
Influence of W/C on plugging capacity.

**Figure 10 materials-17-01568-f010:**
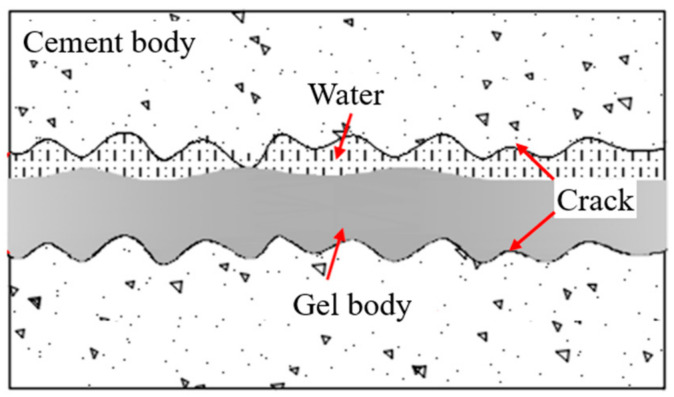
Bleeding effect of cement-based grout.

**Figure 11 materials-17-01568-f011:**
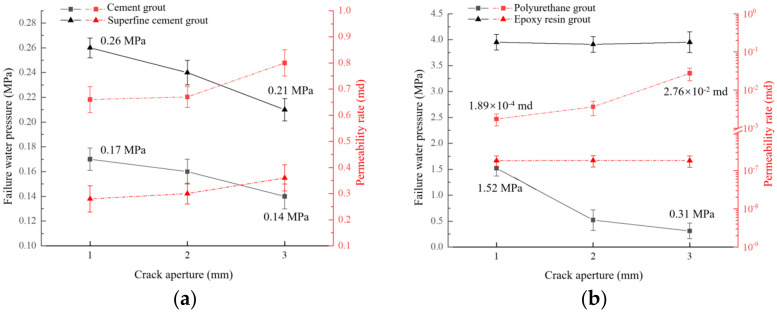
Influence of crack aperture on plugging capacity: (**a**) cement-based grout, grouting pressure = 1 MPa, W/C = 0.8; (**b**) chemical grout, grouting pressure = 2 MPa.

**Figure 12 materials-17-01568-f012:**
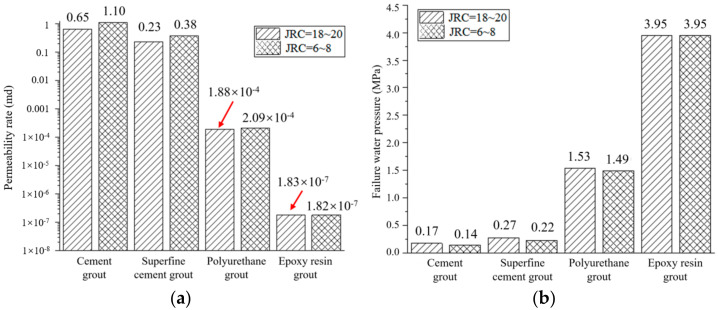
Influence of crack roughness on plugging capacity: (**a**) permeability rate; (**b**) failure water pressure.

**Figure 13 materials-17-01568-f013:**
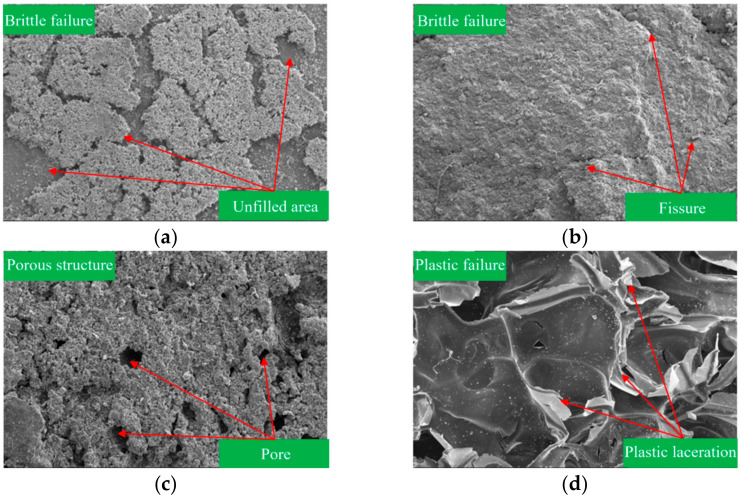
Microstructure magnified 200 times: (**a**) cement gel; (**b**) ultrafine cement gel; (**c**) polyurethane gel; (**d**) epoxy resin gel.

**Figure 14 materials-17-01568-f014:**
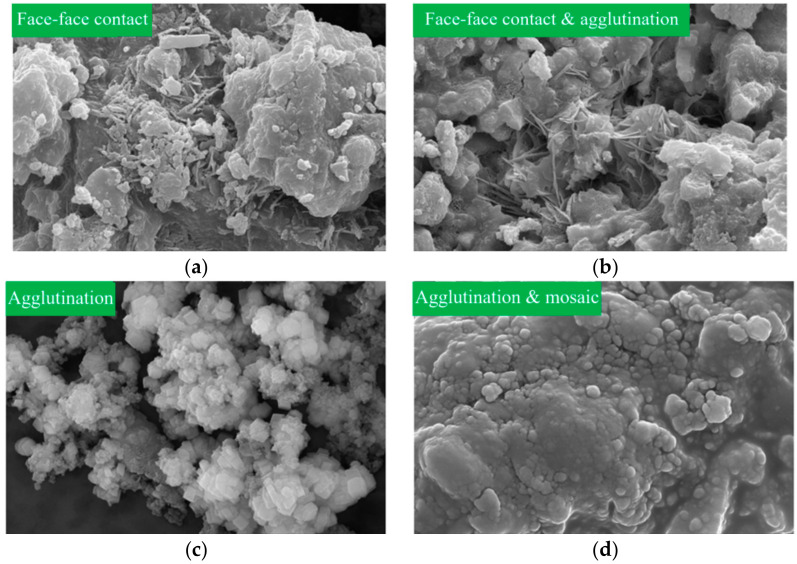
Microstructure magnified 20,000 times: (**a**) cement gel; (**b**) ultrafine cement gel; (**c**) polyurethane gel; (**d**) epoxy resin gel.

**Figure 15 materials-17-01568-f015:**
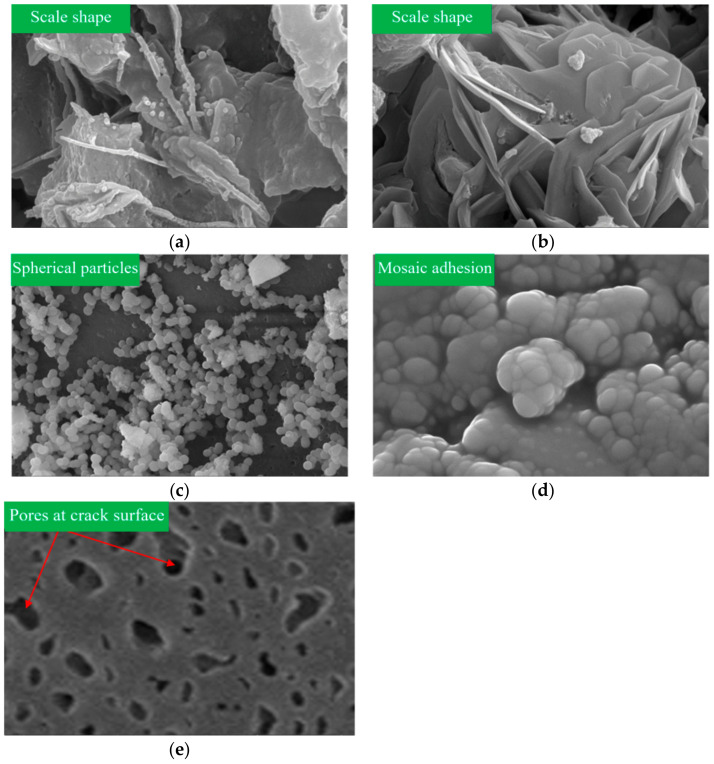
Microstructure magnified 100,000 times: (**a**) cement gel; (**b**) ultrafine cement gel; (**c**) polyurethane gel; (**d**) epoxy resin gel; (**e**) crack surface.

**Figure 16 materials-17-01568-f016:**
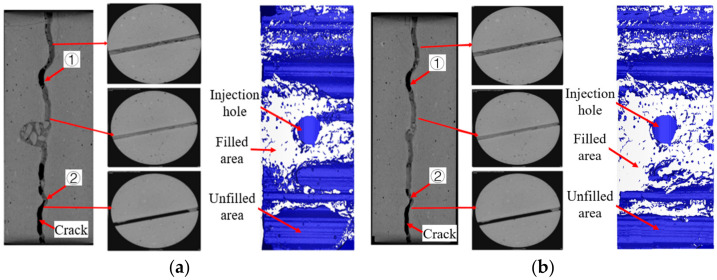
Filling state of cement-based grout gel in crack: (**a**) cement gel; (**b**) ultrafine cement gel.

**Table 1 materials-17-01568-t001:** Parameters of ordinary Portland cement and ultrafine cement (800 mesh).

Material Type	Initial Setting Time (min)	Final Setting Time (min)	Compressive Strength (MPa)		Flexural Strength (MPa)	
			3d	28d	3d	28d
Portland cement	131	217	17	45.2	3.5	6.5
Ultrafine cement	152	423	5	30	60	5

**Table 2 materials-17-01568-t002:** Parameters of polyurethane grout and epoxy resin grout.

Material Type	Initial Setting Time (min)	Final Setting Time (min)	Compressive Strength (MPa)	Viscosity (mPa·s)
Polyurethane grout	Immediate reaction to water	144	0.2	800~1000
Epoxy resin grout	60	1440	80	1000~1400

**Table 3 materials-17-01568-t003:** Test details.

Grout	W/C	Grouting Pressure (MPa)	Crack Aperture (mm)	Crack Roughness: JRC
Ordinary Portland/ultrafine cement paste	0.8	0.5, 1.0, 1.5, 2.0, 2.5, 3.0	1	18~20
	0.6, 0.8, 1.0, 1.2, 1.4	1.0	1	18~20
	0.8	1.0	1, 2, 3	18~20
	0.8	1.0	1	6~8, 18~20
Polyurethane grout	—	0.5, 1.0, 1.5, 2.0, 2.5, 3.0	1	18~20
	—	2.0	1, 2, 3	18~20
	—	2.0	1	6~8, 18~20
Epoxy resin grout	—	0.5, 1.0, 1.5, 2.0, 2.5, 3.0	1	18~20
	—	2.0	1, 2, 3	18~20
	—	2.0	1	6~8, 18~20

## Data Availability

Data are contained within the article.

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
