# Peer review of "Experimental Study on Water-Plugging Performance of Grouted Concrete Crack"

_materials, 2024, doi:10.3390/ma17071568_

Round 1

Reviewer 1 Report

Comments and Suggestions for Authors

This is very interesting research. The work is well-structured, coherent, scientifically sound, and clearly written. However, although the literature review is appropriate, it was unfortunately not used in the analysis of the results.

Therefore, it is necessary to address some corrections and questions, and enrich the discussion of the results with references to previous work.

- Did you use some international standards in your methodology? Include reference(s)

- The JRC profile in Figure 6 is too small

- Some aspects from Table 5 aren’t clear. The first and last rows on each grout have one group of data that apparently are equivalent

- Include the reference and a short description of the Brazilian splitting method

- It is essential to support and contrast the results and their analysis with previous publicized work from other authors. Hence must include several references in section 3 “Results and discussion”.

Author Response

On behalf of my co-authors, we thank you very much for giving us an opportunity to revise our manuscript, we appreciate editor and reviewers very much for their positive and constructive comments and suggestions on our manuscript entitled “Experimental study on water-plugging performance of grouted concrete crack” (ID: Materials-2853822).

We believe that the comments have been highly constructive and very useful to

improve the manuscript. We have studied reviewer’s comments carefully and tried our best to revise our manuscript according to the comments. We appreciate the important points raised by the reviewers and provide a comprehensive revision of the original manuscripts.

Responses to each point raised by the reviewers are listed in the Detailed Response to Reviewers. Attached please find the revised version, which we would like to submit for your kind consideration. We would like to express our great appreciation to you and reviewers for comments on our paper. Looking forward to hearing from you.

Thank you and best regards.

Sincerely,

Zhipeng Li

School of Transportation and Civil Engineering, Shandong Jiaotong University, Jinan, China.

[email protected] / [email protected]

Reviewer 2 Report

Comments and Suggestions for Authors

The article concerns the study of various types of materials enabling repairs of concrete structures using grouting. This is an important practical issue. The authors tested various materials used for this purpose and compared their effectiveness.

Noteworthy is using a test stand that tests various parameters while visually observing the crack behavior. Moreover, the authors used an interesting approach to imitating cracks, allowing their parameterization.

Remarks:

Tables 1 and 2 should be merged, similarly to tables 3 and 4.

Lines 278-289 - poor fluidity of cement grout results from the huge specific surface of ultra-fine cement. The obvious solution is using a superplasticizer, which will simultaneously increase fluidity, reduce w/c ratio, and increase viscosity. All these factors should improve the water-plugging performance.

The text in lines 276-289 is repeated in lines 290-303

Fig. 12 should be placed elsewhere (not immediately after heading 3.3)

Lines 326-332 – is the explanation provided for the described phenomenon supported by research? Perhaps these considerations should be based on research results presented in another publication.

The entry 1mm is incorrect - the unit should be placed after a space, i.e. 1 mm – it applies to the entire manuscript and all units.

We use the notation cement paste rather than cement slurry in cement/concrete technology.

I believe that permeability should be given in SI units rather than in md.

Figure 15 should be on one page or divided into two figures.

Details of the CT analysis should be included in point 2

The paper presents a quantitative comparison of the effectiveness of different types of grouts used to seal cracks. These are research results that are worth publishing. A valuable part of the manuscript is microscopic photos of the examined materials, which allow for a broader analysis and drawing of certain conclusions.

I recommend the work for publication, however, after making appropriate corrections.

Author Response

(The authors gave the same response as above.)
